# MAKING TRANSFORMER DECODERS BETTER DIFFER-ENTIABLE INDEXERS

**Wuchao Li**[1], **Kai Zheng**[2], **Defu Lian**[1],[*] **Qi Liu**[1], **Wentian Bao**[3], **Yun En Yu**[3], **Yang Song**[3],
**Han Li**[2], **Kun Gai**[2]

[1] University of Science and Technology of China,   [2] Kuaishou,   [3] Independent
{liwuchao, qiliu67}@mail.ustc.edu.cn, {kaizheng, lihan08}@kuaishou.com
liandefu@ustc.edu.cn, wb2328@columbia.edu, yuenyun@126.com, ys@sonyis.me
gai.kun@qq.com

## ABSTRACT

Retrieval aims to find the top-k items most relevant to a query/user from a large dataset. Traditional retrieval models represent queries/users and items as embedding vectors and use Approximate Nearest Neighbor (ANN) search for retrieval. Recently, researchers have proposed a generative-based retrieval method that represents items as token sequences and uses a decoder model for autoregressive training. Compared to traditional methods, this approach uses more complex models and integrates index structure during training, leading to better performance. However, these methods remain two-stage processes, where index construction is separate from the retrieval model, limiting the model's overall capacity. Additionally, existing methods construct indices by clustering pre-trained item representations in Euclidean space. However, real-world scenarios are more complex, making this approach less accurate. To address these issues, we propose a Unified framework for Retrieval and Indexing, termed **URI**. URI ensures strong consistency between index construction and the retrieval model, typically a Transformer decoder. URI simultaneously builds the index and trains the decoder, constructing the index through the decoder itself. It no longer relies on one-sided item representations in Euclidean space but constructs the index within the interactive space between queries and items. Experimental comparisons on three real-world datasets show that URI significantly outperforms existing methods.

## 1 INTRODUCTION

Retrieval is a critical upstream stage in various tasks, such as information retrieval and recommender systems, as it sets the upper bound for downstream ranking tasks. Given a query, the retrieval model identifies the top-k most relevant items from a large-scale dataset. Throughout the paper, "query" will be used to represent both query and user.

### 1.1 PRIOR METHODS AND LIMITATIONS

Traditional retrieval models generate representations for queries and items, using simple similarity measures like the inner product. After index construction, the Approximate Nearest Neighbor search retrieves the relevant items. However, this approach has notable performance limitations due to the linearity of the similarity metric, which constrains model complexity, and the decoupling of model training from index construction.

To overcome these limitations, researchers have proposed a new method, Generative Retrieval (Tay et al., 2022; Wang et al., 2022; Rajput et al., 2024; Feng et al., 2022), introducing Transformer decoders (Vaswani, 2017) into the retrieval process. These methods involve multiple stages: first constructing a hierarchical clustering index for items, representing them as sequences of cluster IDs, and then training a decoder to generate these sequences. Compared to traditional methods, these approaches use more complex model architectures and establish connections between model training and the constructed index, resulting in improved performance. However, index construction is still

---

[*]Corresponding author.

disconnected from the retrieval model, relying only on item information without considering queries, and heavily dependent on simplistic similarity measures and clustering algorithms in Euclidean space, greatly limiting index effectiveness.

## 1.2 OUR SOLUTION

In this work, we propose a Unified framework for the Retriever and Indexer, termed **URI**. In URI, the Retriever and Indexer share the same Decoder model, so the index is not constructed using simple similarity measures but is directly generated by the Retriever. In previous methods, the Decoder only memorizes the pre-constructed index, whereas in URI, the Decoder acts as a differentiable index constructor.

We introduce a new metric for generative retrieval indices, called Average Partition Entropy (APE), which is model-independent and can be evaluated immediately after construction. Using this metric, we formally define the index construction task and propose a theoretically effective EM (Dempster et al., 1977) algorithm. We then employ a machine learning approach to simulate the conditions and outcomes of this algorithm by designing loss functions. In previous works, after index construction, an encoder model is only required on the query side to extract information, while the item side is represented by a sequence of index ID tokens. However, in URI, index construction and model training occur simultaneously, so we set encoders on both the query and item sides. Information extracted from the query and item by their respective encoders is input to the Decoder, resulting in a probability distribution over $k$ tokens for each. Our loss functions are trained based on these distributions, enabling the query to identify positive items and the item to learn its assigned token.

## 1.3 OUR CONTRIBUTIONS

- We propose a novel unified framework for generative retrieval, **URI**, which constructs the index simultaneously with training the Transformer Decoder Retriever.

- We introduce a new evaluation metric for generative indices, Average Partition Entropy (APE), which allows direct assessment of index quality immediately after construction, without model training.

- Using APE, we formally define the generative index construction task and provide a theoretical explanation for the rationale behind URI's loss function.

- We conduct extensive experiments on three real-world datasets, showing that URI achieves state-of-the-art performance as a generative retrieval method. Moreover, we design various experiments to explain its superior performance from multiple perspectives.

## 2 RELATED WORK

**Traditional Retrieval**   Traditional retrieval represents both queries and items as embedding vectors, typically measuring their similarity using the inner product, Euclidean distance, or cosine similarity. The method for learning these embeddings varies by task. In document retrieval, pre-trained NLP models like BERT (Kenton & Toutanova, 2019) and ALBERT (Lan, 2019) are commonly used. In recommender systems, dual-tower models are mostly used, including MF (He et al., 2016), Deep-ICF (Xue et al., 2019), and NCF (He et al., 2017) for general recommendations, as well as SASRec (Kang & McAuley, 2018), YouTubeDNN (Covington et al., 2016), and GRU4Rec (Hidasi, 2015) for sequential recommendations. Once trained, various indices are constructed for the item set, including graph-based (Malkov & Yashunin, 2018; Jayaram Subramanya et al., 2019), quantization-based (Jegou et al., 2010), LSH-based (Datar et al., 2004), and tree-based (Bentley, 1975) indices. Approximate Nearest Neighbor Search (ANNS) is then used to complete retrieval.

**Generative Retrieval**   Existing generative retrieval methods use a multi-stage training approach. They first construct a hierarchical clustering index based on pre-trained representations, use cluster-ID sequences to represent items, and then train the model to generate these sequences. DSI (Tay et al., 2022) and NCI (Wang et al., 2022) both construct indices by performing hierarchical K-means clustering on text representations. Additionally, NCI uses a Prefix-Aware Weight-Adaptive Decoder to adapt to different codebook levels. RecForest (Feng et al., 2022) pre-trains a recommendation model and constructs the index using item ID representations, enhancing performance

by integrating multiple trees. TIGER (Rajput et al., 2024) adopts RQ-VAE (Zeghidour et al., 2021) to build the index from item text representations and uses the item's semantic ID sequence to model user historical behavior sequences. DR (Gao et al., 2021) uses an MLP for inference and iteratively updates the index with the EM (Dempster et al., 1977) algorithm.

## 3   A MEASURE OF INDEX: APE

Inspired by the Mutual Information (Dhillon et al., 2003), we propose a metric to measure the index in generative methods. We consider the case where the index has a single layer, meaning the semantic ID length is 1. In later sections, we extend this to an index with $L$ layers. Given a query $x$ corresponding to positive items $\mathcal{Y}_x^+ = \{y^1, \ldots, y^n\}$, the $n$ items are distributed into $k$ buckets after indexing. The query then obtains a probability distribution $Q = [q_1, \ldots, q_k]$ over the $k$ buckets through the generative model's decoder. To the best of our knowledge, all existing work uses cross-entropy loss to ensure that the input query $x$ retrieves buckets containing samples from $\mathcal{Y}^+$. Let $\mathrm{Idx}(\cdot)$ denote the constructed index that maps an item to a specific bucket, with $\mathrm{Idx}(y_i) = b_i$. The loss function is defined as

$$\mathcal{L}_{ce}(x, \mathcal{Y}_x^+) = -\sum_{i=1}^{n} \log q_{b_i} = -\sum_{j=1}^{k} n_j \log q_j, \text{ where } n_j = \sum_{i=1}^{n} I(b_i = j). \tag{1}$$

It can be easily proven that the loss attains its minimum value when $q_j = n_j/n$ using the method of Lagrange multipliers. Substituting $q_j$, we can get

$$\mathcal{L}_{ce}(x, \mathcal{Y}_x^+) \geq -\sum_{j=1}^{k} n_j \log \frac{n_j}{n} = -n \sum_{j=1}^{k} \frac{n_j}{n} \log \frac{n_j}{n} \tag{2}$$

The lower bound of the loss function is established once the index is constructed. It depends only on the concentration of positive items corresponding to a query within the index. Let $\mathcal{Q}_x(\mathrm{Idx}) = [\frac{\sum_i^n I(\mathrm{Idx}(y_i)=1)}{n}, \ldots, \frac{\sum_i^n I(\mathrm{Idx}(y_i)=k)}{n}] = [\frac{n_1}{n}, \ldots, \frac{n_k}{n}]$ represent the frequency distribution of the buckets containing the positive items corresponding to the query $x$. The entropy of $\mathcal{Q}_x$, $H(\mathcal{Q}_x(\mathrm{Idx})) = -\sum_j^k \frac{n_j}{n} \log \frac{n_j}{n}$, significantly affects the retrieval performance of the query $x$ after cross-entropy training. Inspired by this, we propose a new evaluation metric for the index in generative retrieval models, called Average Partition Entropy (APE).

$$APE(\mathrm{Idx}) = \mathbb{E}_{x \in \mathcal{D}}[H(\mathcal{Q}_x(\mathrm{Idx})] \tag{3}$$

APE measures the concentration of relevant items in the index throughout the dataset. It depends solely on the dataset and the constructed index, independent of the neural network model, allowing APE to preliminarily evaluate index quality. A smaller APE value indicates better index construction. Balance is crucial in index construction, and all methods account for it when building indices. The proposed APE metric is meaningful only when the index is relatively balanced. For extremely unbalanced indices, such as when all items are assigned to one bucket, APE reaches its minimum value, but making it meaningless. In generative retrieval, most models use a hierarchical, multi-layer index structure. For an index with $L$ layers and width $k$, we can obtain $\mathcal{Q}_x \in R^{k^L}$ at the leaf level.

## 4   METHODOLOGY

### 4.1   PROBLEM FORMULATION AND MOTIVATION

Before introducing the proposed framework URI, we first formulate the task of index construction in generative retrieval and discuss how to solve it. Given a dataset $\mathcal{D}$ consisting of a query set $\mathcal{X}$ and an item set $\mathcal{Y}$ of size $N$, where each query $x \in \mathcal{X}$ has its corresponding set of positive samples $\mathcal{Y}_x^+ \subset \mathcal{Y}$, the task is to allocate all items evenly into $k$ buckets, with each bucket containing $N/k$ items. The mapping from items to buckets is denoted as $\mathrm{Idx}(\cdot)$. The optimal index is defined as:

$$\arg\min_{\mathrm{Idx}} APE(\mathrm{Idx}) \tag{4}$$

Even allocation is introduced to prevent all items from being assigned to a single bucket, which would result in $APE = 0$ and render it meaningless. Assuming $N$ is divisible by $k$, and $z =$

$N/k$, there are $N!/(k! \cdot z!)$ possible allocation schemes (indices). This is an NP-hard allocation problem, making direct optimization infeasible due to high computational cost. Existing methods attempt to approximate the optimal solution by clustering in the item representation space; however, a significant gap remains. These methods assume that items close in Euclidean distance yield similar model outputs, based on the premise of a small Lipschitz constant. However, as models increase in complexity, the cumulative effect of the Lipschitz constant becomes more pronounced (Fazlyab et al., 2019), challenging this assumption.

To address this issue, we propose constructing the index directly from the final retrieval model, rather than in the shallow representation space. Some previous methods like DR (Gao et al., 2021) and JTM (Zhu et al., 2019) iteratively update the index based on the model after training, using the EM (Dempster et al., 1977) algorithm to improve index quality. This is because the model memorizes the index while capturing the rich collaborative filtering information within the dataset at the same time. In Section 3, we define $\mathcal{Q}_x$ under the ideal assumption of the model's memorization of the index. However, in practice, $\mathcal{Q}_x$ is influenced by complex collaborative filtering relationships between queries and items, preventing convergence to the minimum point of $\mathcal{L}_{ce}(x, \mathcal{Y}_x^+)$. The actual $\mathcal{Q}_x$ contains all the information absorbed by the model during training. Reconstructing the index based on the distribution $\mathcal{Q}_x$ obtained after training allows the new index to outperform the original. We propose a simple greedy algorithm for reconstruction.

**Greedy Algorithm:** For each item $y$ with a relevant query set $\mathcal{X}_y^+ = \{x_1, \ldots, x_n\}$ (for each $x \in \mathcal{X}_y^+$, $y \in \mathcal{Y}_x^+$), we calculate the expected distribution $\mathbb{E}_{x \in \mathcal{X}_y^+}[\mathcal{Q}_x]$ of these queries. The item is assigned to the token with the highest probability in its expected distribution. Under certain conditions, this greedy algorithm can be proven to correctly assign the majority of items.

**Theorem 1.** *Given a dataset $\mathcal{D}$ consisting of $m$ queries and $N$ items, where the $N$ items are evenly allocated into $k$ buckets, and the probability of each query being related to an item is $p$. Given the probability distribution $\{\mathcal{Q}_x\}_{x \in \mathcal{X}}$ of each query across the $k$ buckets, using the aforementioned greedy algorithm can recover the allocation scheme of each item. Given*

$$m \geq \frac{\left[\Phi^{-1}(1-\delta)\right]^2 \left(1 + \frac{2N}{k} p(1+p)\right)}{p(1-p)},$$

*the probability of successfully allocating each item $P_{correct} \geq 1 - \delta$, where $\Phi^{-1}$ is the inverse CDF of the standard normal distribution.*

For a detailed proof, see Appendix D. In real-world datasets, statistical values often meet the theorem's conditions, indicating that this greedy algorithm is reliable. Therefore, the optimal index problem can be optimized using the EM algorithm: (1) At the E-step, the model optimizes $\mathcal{L}_{ce}(x, \mathcal{Y}_x^+)$ based on a pre-constructed index and obtains converged $\mathcal{Q}_x$. (2) At the M-step, the index is reconstructed using the greedy algorithm described above, based on $\mathcal{Q}_x$.

We simulate the M-step using a machine-learning approach. As discussed in Section 3, training with cross-entropy aligns the query distribution with the expected distribution of related items. By swapping the roles of items and queries, the item distribution can be similarly trained using cross-entropy. Considering an item $y$ and its relevant queries $\mathcal{X}_y^+ = \{x^1, \ldots, x^n\}$, for $x^i \in \mathcal{X}_y^+$, let $\mathcal{Q}_{x^i} = [q_1^i, \ldots, q_k^i]$. Given the probability distribution $P_y = [p_1, \ldots, p_k]$ of item $y$ over $k$ tokens from the decoder, the cross-entropy is:

$$\mathcal{L}_{ce}(y, \mathcal{X}_y^+) = -\sum_{i=1}^n \sum_{j=1}^k q_j^i \log p_j = -\sum_{j=1}^k ((\sum_{i=1}^n q_j^i) \log p_j). \tag{5}$$

It is straightforward to prove that the loss function reaches its minimum value when $p_j = (\sum_i^n q_j^i)/n$. This method completes the first step of the greedy algorithm. Next, we assign the item to the token with the highest probability in the distribution. We introduce entropy as the loss function to approximate the item distribution to a one-hot distribution:

$$\mathcal{L}_{commit}(y) = -\sum_{j=1}^k p_j \log p_j. \tag{6}$$

Note that index construction requires a constraint that items must be evenly allocated. To meet this constraint, we also use a machine-learning approach, introducing the expected distribution to

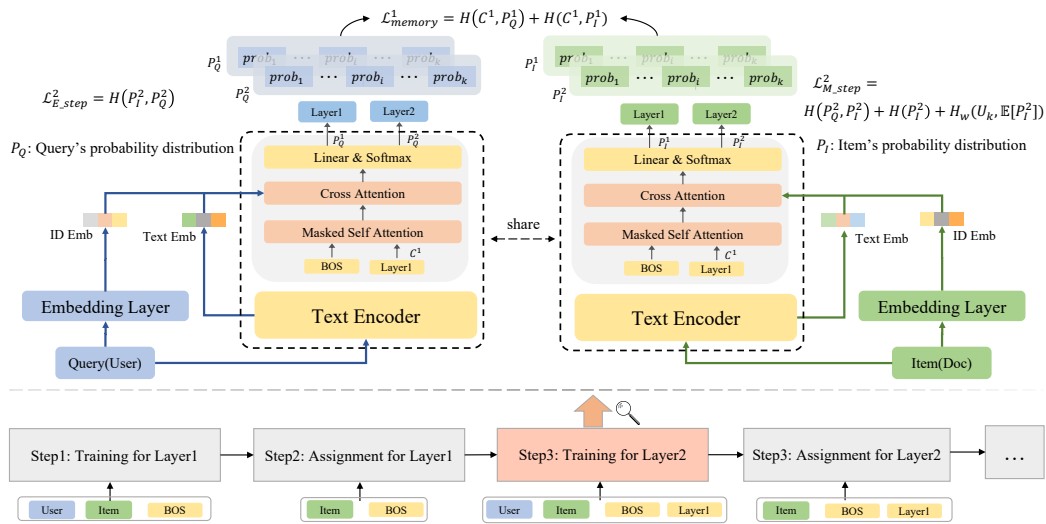

Figure 1: Overview Framework of URI's Retriever and Indexer.

approximate a uniform distribution. The specific form is given by:

$$\mathcal{L}_{balance}(\mathcal{Y}) = H(U^k, \mathbb{E}_{y \in \mathcal{Y}}[P_y]) = -\sum_{j=1}^{k} \frac{1}{k} \log \mathbb{E}_{y \in \mathcal{Y}}[p_j^y]. \tag{7}$$

$U^k$ is a uniform distribution over $k$ tokens. $\mathcal{L}_{commit}$ aims to approximate the distribution to a one-hot distribution, while $\mathcal{L}_{balance}$ aims to approximate the distribution to a uniform distribution. However, these objectives do not conflict, as $\mathcal{L}_{commit}$ acts on individual items, while $\mathcal{L}_{balance}$ acts on the entire item set $\mathcal{Y}$.

## 4.2 DECODER AS INDEXER AND RETRIEVER

Building on the previous discussion of index construction, we propose a unified framework for the retriever and indexer, called URI. URI simulates the above EM algorithm using a differentiable machine-learning approach, enabling end-to-end training. URI uses the same decoder model to generate distributions for both the query and item over $k$ tokens. We denote the distributions at the $l$th token obtained from the decoder for the query and item as $P_Q^l$ and $P_I^l$, respectively. $H(\cdot, \cdot)$ represents cross-entropy, and $H(\cdot)$ represents entropy. The loss functions are summarized as follows:

$$\mathcal{L}_{E\_step}^l = \mathcal{L}_{ce}^l(x, \mathcal{Y}_x^+) = H(P_I^l, P_Q^l) \tag{8}$$

$$\mathcal{L}_{M\_step}^l = \mathcal{L}_{ce}^l(y, \mathcal{X}_y^+) + \mathcal{L}_{commit}^l + \mathcal{L}_{balance}^l = H(P_Q^l, P_I^l) + H(P_I^l) + H(U^k, \mathbb{E}[P_I^l]), \tag{9}$$

Note that the two losses are trained simultaneously; the terms "E-step" and "M-step" are used for clarity. From a gradient optimization perspective, the two $\mathcal{L}_{ce}$ terms ensure that $P_Q$ and $P_I$ of corresponding queries and items are as close as possible. $\mathcal{L}_{commit}$ prevents $P_Q$ and $P_I$ from converging to a uniform distribution, while $\mathcal{L}_{balance}$ prevents them from converging to the same one-hot distribution, avoiding index collapse.

We use positive sample pairs (query, item) to train the two loss functions. However, relying solely on positive samples introduces bias, as items not in the positive sample set cannot be properly learned. To address this, we incorporate randomly sampled items to train $\mathcal{L}_{commit}$ and $\mathcal{L}_{balance}$. These random items do not participate in optimizing $\mathcal{L}_{ce}$, as they lack corresponding matching queries. In each training batch, we sample a number of random items equal to the positive samples. Furthermore, $\mathcal{L}_{balance}$ lacks adaptability across different tokens. To mitigate this, we introduce weighting mechanisms to adjust the distribution across tokens:

$$\mathcal{L}_{balance}^l = H_w(U^k, \mathbb{E}[P_I^l]) = -\sum_{j=1}^{k} \text{sgd}(w_j) \cdot \frac{1}{k} \log \mathbb{E}_{y \in \mathcal{Y}}[p_j^y], \; w_j = |\frac{1}{k} - \log \mathbb{E}_{y \in \mathcal{Y}}[p_j^y]| + 1 \tag{10}$$

The weight's magnitude relates to the difference between $\mathbb{E}[P_I^l]$ and the uniform distribution. A larger difference indicates a higher imbalance, requiring a larger weight. A constant factor of 1 ensures weight stability, preventing it from approaching zero.

**Assignment** The Transformer Decoder uses autoregressive training, meaning that when training the next layer's token, previous tokens must be input. In our approach, since each layer's tokens are unknown before training, we train the model layer by layer. Once the model converges, items are explicitly assigned to $k$ tokens, preparing for training the next layer. To assign items, we first pass all items through the Decoder to obtain each item's probability distribution. Based on this probability distribution, we use a balanced assignment algorithm (Li et al., 2023). Specifically, we set a maximum load for each token, typically the average allocation size $N/k$. The probability distribution of each item is sorted in descending order, and tokens are selected sequentially. If the number of items in a selected token reaches the maximum load, the next token is chosen.

During training, we first train the model on the first layer, optimizing $\mathcal{L}_{E\_step}^1 + \mathcal{L}_{M\_step}^1$. Once the model converges, we assign first-layer tokens to the items. Next, we train the second-layer tokens, using the first-layer tokens assigned to the items as inputs to the decoder. To prevent the model from forgetting information learned in the first layer, we introduce a memory reinforcement loss, which is the same as the loss used in prior works for training the decoder after index construction:

$$\mathcal{L}_{memory}^l = -\sum_{i=1}^{l}(H(C^i, P_Q^i) + H(C^i, P_I^i)); \quad \mathcal{L}_{Decoder}^l = \mathcal{L}_{E\_step}^l + \mathcal{L}_{M\_step}^l + \mathcal{L}_{memory}^{l-1} \quad (11)$$

where $C^i$ denotes the one-hot label of the token assigned to the item at layer $i$. Let $\mathcal{L}_{memory}^0 = 0$. We train $\mathcal{L}_{Decoder}^l$ layer by layer from $l = 1$ to $l = L$, where $L$ is a hyperparameter.

Figure 1 shows the training process at Layer 2. The figure illustrates the URI framework, with its core being the decoder component, serving as both the Retriever and the Indexer. Different encoders can be used for different types of queries and items. For example, in a sequential recommendation scenario, the query part can use a sequential model as its encoder. Additionally, features extracted by the encoder are input to the decoder sequentially, allowing any desired representations to be freely added, enriching the index construction with more information.

Although URI requires layer-by-layer training, the number of nodes increases exponentially. Therefore, in practice, training two to three layers is generally sufficient. Moreover, tokens learned using this method are more appropriately named, and their sequences exhibit greater consistency across different layers. This occurs because when learning tokens of later layers, all assigned tokens from preceding layers are used as inputs. Consequently, the information represented by the same token across layers becomes more consistent, enhancing model performance. We validate this through experiments (Sec 5.3).

### 4.3 RANKER

After the training of the Retriever, a ranker will be trained to identify the topk items for the final retrieval. In previous work, most methods (Tay et al., 2022; Wang et al., 2022; Rajput et al., 2024; Feng et al., 2022) involved randomly adding a new token to different items in the final layer to serve as the ranker, while the DR(Gao et al., 2021) model utilizes softmax loss over the entire item set. We follow the approach of DR, taking the inner product of query and item as the score, using sampled softmax loss (Covington et al., 2016) for training, where 10 negative samples are drawn for each positive example. Furthermore, we incorporate hard negative mining. Specifically, we first sample 9 random negative examples, then select one hard negative sample from the sibling nodes of the positive sample's node. This is feasible due to the high quality of the index we construct. We summarize the ranker structure in Figure2.

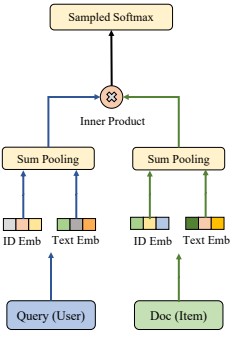

Figure 2: Ranker.

## 4.4 INFERENCE

In the index, each item is represented by a sequence of token IDs, transforming it into a sequence generation task. The Transformer Decoder employs beam search to complete this task, ultimately generating the top-b sequences. These top-b sequences represent the top-b buckets in the final index layer. The union of all items in these buckets forms the candidate set. The final top-k retrieval results are obtained by computing the inner product between the query's representation and item representations in the candidate set.

## 4.5 COMPLEXITY ANALYSIS

Let the time for a single pass through the Decoder be $O(t)$, the dataset size be $|\mathcal{D}|$, and the number of index layers be $L$. Then, the training complexity of URI is $O(2L|\mathcal{D}|t)$. Compared to other methods, URI introduces an additional constant factor of $2L$ due to layer-by-layer training, with each training step involving separate forward passes for both the query and item. This may slightly increase training time; however, this overhead is acceptable, as $L$ is typically small in practice, and URI does not require index construction before training, eliminating that overhead. During inference, URI's complexity is the same as existing methods, dominated by the Decoder with a complexity of $O(BL^3)$, where $B$ is the beam size and $L^3$ is the complexity required for the Decoder to generate a sequence of length $L$.

## 5 EXPERIMENT

We conduct extensive experiments to answer the following questions:

**RQ1** What is the overall performance of **URI** compared to other Generative methods?

**RQ2** Why does **URI** perform so well?

**RQ3** How to set the hyper-parameters for **URI** index?

**RQ4** How is the index progressively built during **URI** training?

## 5.1 EXPERIMENTAL SETTINGS

**Datasets** We select three widely-used real-world datasets, each corresponding to one of three retrieval tasks: document retrieval, general recommendation, and sequential recommendation. The KuaiSAR (Sun et al., 2023) dataset is a short video search dataset, where the query is text and the item is a short video with a text description, corresponding to the document retrieval task. The Beauty and Toys and Games datasets are both from the Amazon platform (He & McAuley, 2016). We use the Beauty dataset for the general recommendation task and the Toys and Games dataset for the sequential recommendation task, where the sequence length is set to 20.

**Baselines and Settings** In this paper, we focus on demonstrating that our method is the optimal generative retrieval approach. To that end, we selected five representative works in this field: DSI (Tay et al., 2022), NCI (Wang et al., 2022), RecForest (Feng et al., 2022), TIGER (Rajput et al., 2024), and DR(Gao et al., 2021). Among these, DSI and NCI are designed for document retrieval tasks, while the other three address recommendation tasks. All five methods follow a multi-stage approach, where the index is first constructed, followed by model training. To ensure fairness in comparison, we maintain consistency with the original papers in terms of the retrieval model architecture and index construction methods. For text extraction, we utilize the 'bert-base-cased' (Kenton & Toutanova, 2019) model from Huggingface (Jain, 2022) across all methods. We set the representation dimension to 96, the number of layers in the Decoder model to 2, the beam size to 20, and the learning rate to 0.001. For NCI and DSI, we set the number of tokens in the final layer to 100, referred to as $c$ in the original papers. For KuaiSAR dataset, we set the index width $k = 64$ and depth $L = 2$. For Beauty and Toys datasets, we set $k = 32$ and $L = 2$.

**Evaluation Metrics** Following most prior works, we evaluate the performance of each method using Recall@K and NDCG@K (where K = 10 or 20). Additionally, we use our newly proposed APE metric to assess the effectiveness of the indices constructed by different methods.

## 5.2 OVERALL PERFORMANCE

Table 1: Recall(R)@K and NDCG(N)@K Results of URI compared with baseline methods

| Dataset | KuaiSAR | | | | Amazon Beauty | | | | Toys and Games | | | |
|---|---|---|---|---|---|---|---|---|---|---|---|---|
| Metric | R@10 | N@10 | R@20 | N@20 | R@10 | N@10 | R@20 | N@20 | R@10 | N@10 | R@20 | N@20 |
| RecForest | 0.0384 | 0.0194 | 0.0626 | 0.0254 | 0.0102 | 0.0046 | 0.0179 | 0.0065 | 0.0344 | 0.0254 | 0.0399 | 0.0267 |
| w/ URI Index | 0.0546 | 0.0270 | 0.0785 | 0.0330 | 0.0129 | 0.0062 | 0.0214 | 0.0084 | 0.0518 | 0.0371 | 0.0586 | 0.0381 |
| DSI | 0.0976 | 0.0519 | 0.1459 | 0.0641 | 0.0105 | 0.0050 | 0.0176 | 0.0068 | 0.0486 | 0.0345 | 0.0536 | 0.0358 |
| w/ URI Index | 0.1027 | 0.0560 | 0.1591 | 0.0700 | 0.0125 | 0.0058 | 0.0216 | 0.0081 | 0.0518 | 0.0373 | 0.0579 | 0.0391 |
| NCI | 0.0568 | 0.0276 | 0.1053 | 0.0397 | 0.0061 | 0.0028 | 0.0110 | 0.0040 | 0.0325 | 0.0206 | 0.0398 | 0.0225 |
| w/ URI Index | 0.0650 | 0.0312 | 0.1174 | 0.0444 | 0.0079 | 0.0037 | 0.0141 | 0.0052 | 0.0362 | 0.0220 | 0.0448 | 0.0242 |
| TIGER | 0.0376 | 0.0187 | 0.0624 | 0.0250 | 0.0092 | 0.0041 | 0.0167 | 0.0058 | 0.0322 | 0.0201 | 0.0383 | 0.0231 |
| w/ URI Index | 0.0546 | 0.0270 | 0.0785 | 0.0330 | 0.0129 | 0.0062 | 0.0214 | 0.0084 | 0.0509 | 0.0360 | 0.0582 | 0.0373 |
| DR | 0.0604 | 0.0358 | 0.0829 | 0.0415 | 0.0105 | 0.0049 | 0.0190 | 0.0075 | 0.0429 | 0.0271 | 0.0478 | 0.0295 |
| w/ URI index | 0.1124 | 0.0589 | 0.1533 | 0.0692 | 0.0125 | 0.0058 | 0.0203 | 0.0080 | 0.0523 | 0.0312 | 0.0607 | 0.0384 |
| URI | **0.1793** | **0.0925** | **0.2594** | **0.1127** | **0.0157** | **0.0067** | **0.0294** | **0.0097** | **0.0573** | **0.0373** | **0.0685** | **0.0401** |

In this section, we address **RQ1** by comparing the overall performance of URI against baseline methods using Recall@10, Recall@20, NDCG@10, and NDCG@20. Additionally, we replace all baselines' original indices with the index constructed by URI while keeping their model structures unchanged. Table 1 summarizes all results, from which we make the following observations.

URI outperforms all generative retrieval methods significantly across all datasets. For Recall@10, URI improves upon the best baseline by 83.7%, 49.5%, and 33.6% on the KuaiSAR, Beauty, and Toys and Games datasets, respectively. On the KuaiSAR dataset, URI achieves over a 100% improvement compared to most baselines. This is because the text in the dataset is token-anonymized, preventing the direct use of pre-trained large language models for representation extraction. Consequently, we train a small language model from scratch on the limited data, restricting the effectiveness of indices built with these representations and resulting in poorer baseline performance. However, URI significantly mitigates this issue by learning query-item correlations, demonstrating the potential of constructing indices directly from the interaction space of queries and items. Additionally, when all baselines use the index constructed by URI, performance improves to varying degrees, indicating URI's index is highly effective across different model structures.

## 5.3 EFFECTIVENESS ANALYSIS

We analyze the reasons behind URI's performance from multiple perspectives to answer **RQ2**.

**APE Comparison** We evaluated the APE of indices constructed by different methods, including Random, K-means, VQ-VAE (Van Den Oord et al., 2017), RQ-VAE (Zeghidour et al., 2021), and URI. For each dataset, we select different values of $k$ under a typical two-layer setting, allowing us to compare indices with varying partition granularities. Since the KuaiSAR dataset has a larger number of items, we chose relatively larger values of $k$. All results are summarized in Table 2. As shown in the table, the APE values of the indices constructed by URI are significantly lower than those of other methods, indicating that related items for the same query are more tightly clustered. This leads to more precise retrieval performance after training.

**Token Consistency** In previous index construction methods, clusters are named randomly. For multi-layer indices, this approach has little impact on the first layer. However, for subsequent layers, relationships between clusters at different levels are not considered during naming, and tokens are randomly assigned, limiting the index's effectiveness. In URI, layer-by-layer training allows information from all previously constructed tokens to be incorporated when constructing the current layer. Consequently, during naming, similar clusters are assigned the same token. To demonstrate this, we design two variants of the URI index, in which token naming is degraded.

- *Variant 1*: A uniform token transformation is applied to all tokens in the final layer, meaning transformations are the same for nodes under different parent nodes.
- *Variant 2*: A transformation is applied to tokens in the final layer, but transformations differ for nodes under different parent nodes.

Table 2: Comparison of the APE values for indices constructed by different methods.

| Dataset | $k$ | Random | Kmeans | VQ-VAE | RQ-VAE | URI |
|---|---|---|---|---|---|---|
| KuaiSAR | 32 | 1.222 | 0.915 | 0.922 | 0.909 | **0.654** |
| | 64 | 1.224 | 1.037 | 1.042 | 1.032 | **0.954** |
| | 128 | 1.224 | 1.141 | 1.144 | 1.143 | **1.135** |
| Beauty | 16 | 1.127 | 1.080 | 1.088 | 1.075 | **0.988** |
| | 32 | 1.134 | 1.106 | 1.112 | 1.110 | **1.040** |
| | 64 | 1.135 | 1.125 | 1.128 | 1.126 | **1.107** |
| Toys and Games | 16 | 1.756 | 1.662 | 1.672 | 1.662 | **1.594** |
| | 32 | 1.770 | 1.729 | 1.733 | 1.728 | **1.660** |
| | 64 | 1.771 | 1.763 | 1.765 | 1.761 | **1.728** |

Figure 5 illustrates these two variants. Note that both variants retain the same APE as the original URI index. Table 3 presents the Recall@10 results. Compared to the original index, the poorer results of *Variant 1* suggest that the original index maintains consistency across clusters with the same token across different layers. Furthermore, the worse performance of *Variant 2* compared to *Variant 1* indicates that, within the same layer, token naming for nodes under different parent nodes also exhibits consistency.

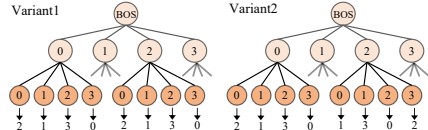

Figure 3: Variant Illustration.

Table 3: Recall@10 of URI and variants

| Dataset | KuaiSAR | Beauty | Toys and Games |
|---|---|---|---|
| Variant 1 | 0.1633 | 0.0138 | 0.0561 |
| Variant 2 | 0.1581 | 0.0124 | 0.0519 |
| Original URI | 0.1793 | 0.0157 | 0.0573 |

**Adaptive Weight in $\mathcal{L}_{balance\_w}$**   In Equation 10, we propose adaptive weight loss $\mathcal{L}_{balance}$. Here, we demonstrate its necessity through experiments. Table 4 shows that removing the adaptive weight significantly declines URI's performance. When training with the original $\mathcal{L}_{balance}$, many tokens are ignored, resulting in an "empty bucket" state. Introducing adaptive weight ensures the optimizer focuses on these buckets.

Table 4: Ablation results of Recall@10 for Adaptive Weight and Hard Negative Mining

| Dataset | KuaiSAR | Beauty | Toys and Games |
|---|---|---|---|
| w/o Adaptive Weight | 0.1301 | 0.0122 | 0.0517 |
| w/o Hard Negative Mining | 0.1528 | 0.0143 | 0.0558 |
| Original URI | 0.1793 | 0.0157 | 0.0573 |

**Hard Negative Mining**   In Section 4.3, we propose incorporating hard negative samples during ranker training by sampling from the sibling nodes of the positive sample's node. The Recall@10 results in Table 4 indicate that removing hard negative samples decreases URI's overall performance, demonstrating the effectiveness of hard negative sampling. This also provides evidence of the exploitable similarity among sibling nodes within the URI index.

## 5.4   Hyper-Parameter Sensitivity Analysis

In this section, we compare results under different index hyperparameter settings to address **RQ-3**. For generative retrieval models, a larger $k$ implies more parameters and leaf nodes at the same depth $L$, resulting in finer-grained item partitioning and more possible item combinations during retrieval, thus increasing the model's upper performance bound. However, finer partitioning reduces the number of items within the same node, increasing the risk of overfitting. Therefore, it is crucial to select $k$ and $L$ to balance accuracy and robustness in the index. Figure 4 shows the Recall@10 curves as the candidate set size varies under different $k$ and $L$ settings for the three datasets. The

results indicate that $k = 64$ and $L = 2$ are optimal for KuaiSAR, while $k = 32$ and $L = 2$ are most suitable for Beauty and Toys and Games. Under these settings, each leaf node contains approximately 10 to 100 items. In practice, this conclusion can guide the selection of $k$ and $L$.

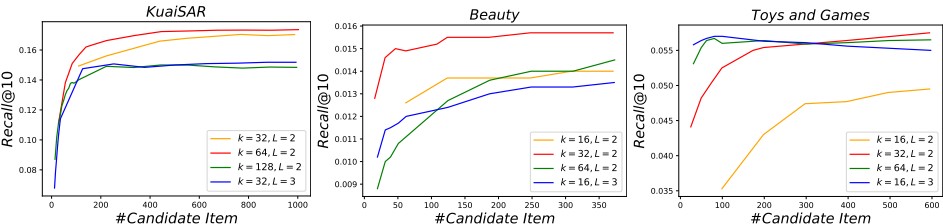

Figure 4: The Recall@10 curve with respect to the number of candidate items on different indices.

## 5.5 CASE STUDY

In this section, we address **RQ4** by observing how the index is constructed, monitoring the token training process of KuaiSAR in the first layer with $k = 4$. We plot the output probability distributions of a specific query (with search_session_id: 2) and its relevant positive items as pie charts after model processing. Figure 5 shows that all probability distributions are initially uniform, indicating that the model has no prior knowledge. As training progresses, the item distributions gradually converge to one-hot. Finally, Item1, Item2, Item4, and Item5 are assigned to Token '4', while Item3 is assigned to Token '2'. The query's distribution closely aligns with the expected value of these five samples' distributions, consistent with the discussion in Section 3. In the original dataset, Item1, Item2, Item4, and Item5 belong to the category "Two-dimensional," while Item3 belongs to "Film and Short Series." This information is not introduced during training, indicating that the model learns similar knowledge solely through collaborative filtering signals between the query and items.

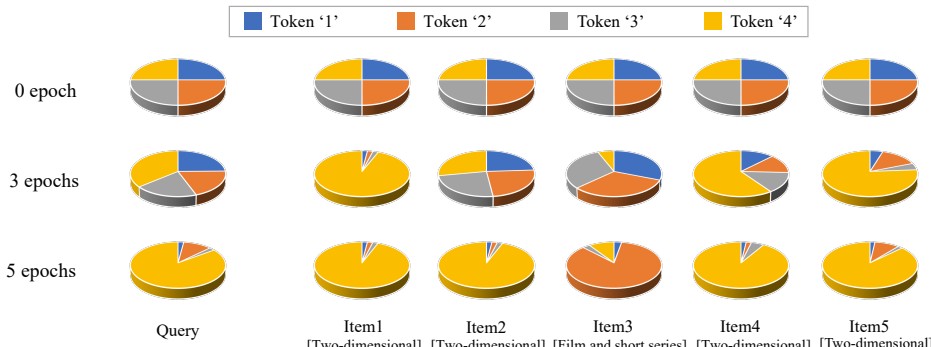

Figure 5: Case Study.

## 6 CONCLUSION AND FUTURE WORK

In this paper, we propose Averaged Partition Entropy (APE), a new metric for evaluating indices in generative models, and introduce a novel generative retrieval framework, URI, where model training and index construction are completed simultaneously within the same Decoder. Compared to previous methods, where the Decoder only memorizes an already constructed index, URI treats the Decoder as a differentiable Indexer. Comparative experiments with baselines show that URI achieves state-of-the-art (SOTA) performance. Analytical experiments explain why URI outperforms from various perspectives, including a lower APE for the index and token consistency across layers. In the future, we will focus on developing methods to enable URI to generate multi-level tokens simultaneously to accelerate training.

ACKNOWLEDGMENTS

The work was supported by grants from the National Natural Science Foundation of China (No. U24A20253).

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

APPENDIX

## A NOTATIONS

All notations in the main text and their descriptions are summarized in Table 5.

| Notation | Description |
|:---:|:---:|
| $x, \mathcal{X}$ | Query, Query set |
| $y, \mathcal{Y}$ | Item, Item Set |
| $\mathcal{X}_y^+$ | Queries related to item $y$ |
| $\mathcal{Y}_x^+$ | Items related to query $x$ |
| $\mathcal{D}$ | A Dataset consisting of $\mathcal{X}$ and $\mathcal{Y}$ |
| $\mathcal{Q}_x$ | The distribution of items related to $x$ in the index |
| $p$ | Item-query relevance probability |
| $m$ | The number of queries |
| $N$ | The number of items |
| $\delta$ | A tolerance level |
| $\Phi(\cdot)$ | The CDF of standard normal distribution |
| $P_Q^i$ | The distribution at the $i$-th token of the query |
| $P_I^i$ | The distribution at the $i$-th token of the item |
| $C^i$ | The token assigned to the item at the $i$-th Layer |
| $H(a, b)$ | Cross-entropy of $a$ and $b$ |
| $H(b)$ | Entropy of $b$ |
| $U^k$ | A uniform distribution over $k$ classes |
| $k$ | The width of the index (The number of tokens) |
| $L$ | The depth of the index (The number of Layers) |

Table 5: Notations

## B DATASET STATISTICS

| Dataset | #Query | #Item | # Interaction |
|:---:|:---:|:---:|:---:|
| KuaiSAR | 191330 | 112388 | 1093920 |
| Beauty | 41895 | 15817 | 162713 |
| Toys and Games | 13271 | 25357 | 90557 |

Table 6: Dataset Statistics

We summarize the statistics of the three datasets in Table 6. Setting the tolerance level $\delta$ to 0.95 and substituting the dataset density for $p$, all three datasets satisfy the conditions of Theorem 1. This indicates that our proposed greedy algorithm is applicable to most datasets and demonstrates the rationality of URI's approach in simulating the greedy algorithm through machine learning.

## C SUPPLEMENT

## D PROOF OF THEOREM 1

Before deriving Theorem 1, we can transform the problem of recovering the allocation scheme into a matrix equation-solving problem. Let $W$ be the interaction matrix between queries and items, with a size of $m \times N$. The value at a given position is 1 if the query is related to the item, and 0

otherwise. In a recommender system, $W$ represents the user purchase matrix. Let $H$ be the item allocation matrix, with a size of $N \times k$, where each row contains exactly one 1, and the sum of each column is $N/k$. Define $M = W \times H$, where $M$ has a size of $m \times k$, representing the frequency distribution of each query over the $k$ buckets. Thus, the problem is transformed into solving for $H$ given $M$ and $W$. The conclusions of the theorem and the greedy algorithm are also correspondingly transformed, as described below.

## D.1   THEOREM

**Theorem 2.** *(Correct Reconstruction Probability)*

*Given the following conditions:*

- *Tolerance Level: A tolerance level $\delta \in (0, 1)$.*

- *Parameters: Positive integers $N$ and $k$, and a probability $p \in (0, 1)$.*

- *Random Matrix $W$: An $m \times N$ binary matrix where each element $W_{r,l}$ is independently and identically distributed (i.i.d.), satisfying:*
$$P(W_{r,l} = 1) = p, \quad P(W_{r,l} = 0) = 1 - p$$

- *Structured Matrix $H$: An $N \times k$ binary matrix satisfying:*

  - *Row Constraint: Each row contains exactly one 1, and all other elements are 0.*
$$\sum_{j=1}^{k} H_{i,j} = 1, \quad \forall i = 1, 2, \ldots, N$$

  - *Column Constraint: Each column contains exactly $N_j = \frac{N}{k}$ ones (assuming $N$ is divisible by $k$).*
$$\sum_{i=1}^{N} H_{i,j} = N_j, \quad \forall j = 1, 2, \ldots, k$$

- *Product Matrix $M$: Defined as $M = W \cdot H$.*

*If the number of rows $m$ in matrix $W$ satisfies:*
$$m \geq \frac{\left[\Phi^{-1}(1 - \delta)\right]^2 \left(1 + \frac{2N}{k} p(1 + p)\right)}{p(1 - p)}$$

*Then, using the following greedy algorithm, the probability that each row of matrix $H$ is correctly reconstructed is:*
$$P_{correct} \geq 1 - \delta$$

## D.2   GREEDY ALGORITHM DESCRIPTION

For each row index $i = 1, 2, \ldots, N$:

1. Identify Set $S_i$: Find all rows in matrix $W$ where the $i$-th column is 1.
$$S_i = \{r \mid W_{r,i} = 1\}$$

2. Compute Vector $s_i$: For each column $j = 1, 2, \ldots, k$, calculate the sum of corresponding entries in matrix $M$:
$$s_i^{(j)} = \sum_{r \in S_i} M_{r,j}$$

3. Select Maximum: Determine the column $j^*$ with the highest sum:
$$j^* = \arg\max_j s_i^{(j)}$$

4. Update Matrix $H$: Assign the 1 in the $i$-th row of $H$ to column $j^*$:
$$H_{i,j^*} = 1$$

### D.3 PROOF

For a given row index $i$ and any incorrect column $j \neq j^*$ (where $j^*$ is the correct column), define the difference:

$$\Delta s_i^{(j)} = s_i^{(j^*)} - s_i^{(j)}$$

Using the structure of the problem:

$$\Delta s_i^{(j)} = c_{i,i} + \sum_{l \neq i}(H_{l,j^*} - H_{l,j})c_{i,l}$$

where: $c_{i,i}$ is the number of rows in matrix $W$ where column $i$ has a value of 1. $c_{i,l}$ is the number of rows in matrix $W$ where both columns $i$ and $l$ have a value of 1.

(a) For $c_{i,i}$:

Since $c_{i,i}$ counts the number of times the $i$-th column in $W$ contains a 1, it can be written as:

$$c_{i,i} = \sum_{r=1}^{m} W_{r,i}$$

where $W_{r,i}$ is an independent Bernoulli random variable with probability $p$. Therefore:

- Expectation:

$$E[c_{i,i}] = m \cdot p$$

- Variance:

$$\text{Var}[c_{i,i}] = m \cdot p(1 - p)$$

(b) For $c_{i,l}$ (when $l \neq i$):

Similarly, $c_{i,l}$ counts the number of times both columns $i$ and $l$ in matrix $W$ contain a 1. Since $W_{r,i}$ and $W_{r,l}$ are independent Bernoulli variables:

- Expectation:

$$E[c_{i,l}] = m \cdot p^2$$

- Variance:

$$\text{Var}[c_{i,l}] = m \cdot p^2(1 - p^2)$$

We now calculate the expectation and variance of $\Delta s_i^{(j)}$.

(a) Expectation of $\Delta s_i^{(j)}$:

Using the linearity of expectation and the fact that the expectation of $H_{l,j^*} - H_{l,j}$ is zero (as both are binary variables):

$$E[\Delta s_i^{(j)}] = E[c_{i,i}] = m \cdot p(1 - p)$$

(b) Variance of $\Delta s_i^{(j)}$:

Since $c_{i,i}$ and $c_{i,l}$ are independent, the variance of $\Delta s_i^{(j)}$ is the sum of the variances of the terms:

$$\text{Var}[\Delta s_i^{(j)}] = \text{Var}[c_{i,i}] + \sum_{l \neq i}(H_{l,j^*} - H_{l,j})^2 \cdot \text{Var}[c_{i,l}]$$

Since $(H_{l,j^*} - H_{l,j})^2$ takes values of 0 or 1 (with 1 occurring when one column contains a 1 and the other does not), let $n$ be the number of times $(H_{l,j^*} - H_{l,j})^2 = 1$. We have:

$$n \leq \left(\frac{N}{k} - 1\right) + \frac{N}{k} \leq \frac{2N}{k}$$

Thus, the variance becomes:

$$\text{Var}[\Delta s_i^{(j)}] = \text{Var}[c_{i,i}] + n \cdot \text{Var}[c_{i,l}]$$

Substitute the values of $\text{Var}[c_{i,i}]$ and $\text{Var}[c_{i,l}]$:

$$\text{Var}[\Delta s_i^{(j)}] = m \cdot p(1-p) + n \cdot m \cdot p^2(1-p^2)$$

By the Central Limit Theorem, since $m$ is a fairly large number, we approximate $\Delta s_i^{(j)}$ as a normally distributed variable:

$$\Delta s_i^{(j)} \sim \mathcal{N}\left(E[\Delta s_i^{(j)}], \text{Var}[\Delta s_i^{(j)}]\right)$$

Thus, the probability of correct assignment is:

$$P\left(\Delta s_i^{(j)} > 0\right) = \Phi\left(\frac{E[\Delta s_i^{(j)}]}{\sqrt{\text{Var}[\Delta s_i^{(j)}]}}\right)$$

where $\Phi$ is the CDF of the standard normal distribution.

Substitute the expressions for $E[\Delta s_i^{(j)}]$ and $\text{Var}[\Delta s_i^{(j)}]$ into the probability formula:

$$P\left(\Delta s_i^{(j)} > 0\right) = \Phi\left(\frac{mp(1-p)}{\sqrt{mp(1-p) + nmp^2(1-p^2)}}\right)$$

Simplify the Expression:

$$P\left(\Delta s_i^{(j)} > 0\right) = \Phi\left(\frac{\sqrt{mp(1-p)}}{\sqrt{1 + np(1+p)}}\right)$$

Substitute $n \leq \frac{2N}{k}, m \geq \frac{\left[\Phi^{-1}(1-\delta)\right]^2 \left(1 + \frac{2N}{k}p(1+p)\right)}{p(1-p)}$:

$$P\left(\Delta s_i^{(j)} > 0\right) = \Phi\left(\frac{\sqrt{mp(1-p)}}{\sqrt{1 + np(1+p)}}\right) \geq 1 - \delta$$

