# OpenReview forum: "Making Transformer Decoders Better Differentiable Indexers"
_ICLR.cc/2025/Conference — ICLR 2025 Poster_

### Official Review · Reviewer_YJMb · 2024-11-02

**Soundness:** 2
**Presentation:** 2
**Contribution:** 2
**Rating:** 6
**Confidence:** 3

**Summary:**

This paper proposes an end-to-end retriever and index, It simultaneously clusters query and relevant items and ensure that they both follow similar distribution over cluster nodes at each level. The URI (proposed approach) ensures that clusters are balanced and accurate.

**Strengths:**

This paper proposes an end-to-end retriever and indexer.
Results are state of the art

**Weaknesses:**

- This paper looks a lot like EHI: End-to-end Learning of Hierarchical Index for Efficient Dense Retrieval with similar loss functions, TMLR 2024
- Results on standard bier benchmark datasets are missing
- In real world scenario new items are always popping in and out of existence, how does URI deals with zero shot items

**Questions:**

Please see the weakness section and answer the following:
- How does method compares with EHI, what are the similarity and differences between EHI and URI
- APE is standard practice to measure cluster quality why is it a contribution in your paper. (See ECLARE)
- Add comparison on BIER benchmark datasets

ECLARE: Extreme Classification with Label Graph Correlations, WWW 2021.

---

> ### Author Response · Authors · 2024-11-21
> **Responses to Reviewer YJMb**
>
> Many thanks for your constructive and detailed comments. We believe that your comments will make our work more robust and convincing. The detailed responses to your comments are listed as follows:
>
> **Weaknesses1:** This paper looks a lot like EHI. What are the similarities and differences?
>
> **Response:** The similarity is that both EHI and URI utilize a unified retriever and indexer and are trained end-to-end. However, there are several key differences:
>
> 1. The two methods use different paradigms for retrieval. URI employs a generative approach to predict the next token, while EHI uses a discriminative approach to predict the relevant nodes at the current layer of the tree index.
> 2. URI employs Transformer Decoder model with causal masks which makes the hierarchical retrieval more effective, whereas EHI uses a simple MLP.
> 3. The design of the loss functions differs; URI's loss function simulates the EM algorithm, merging the two stages into one and does not involve negative sampling, while EHI's loss function primarily relies on contrastive loss between positive and negative samples.
>
> The original EHI paper does not seem to provide code. We attempted to reproduce the EHI algorithm, but found that the loss function was difficult to converge. Due to the lack of explicit balancing constraints, the index easily collapses. We will cite this paper in the camera-ready version and provide a detailed comparison of the two algorithms.
>
> **Weaknesses2:** Results on standard beir benchmark datasets are missing
>
> **Response:** In our paper, we selected three datasets to demonstrate the applicability of URI in the fields of Information Retrieval, General Recommendation, and Sequential Recommendation. We will now include the NQ320K dataset from the BEIR benchmark. The results are summarized in the table below. From the table, we can see that URI outperforms other methods on NQ320K dataset.
>
> |  Method   | Recforest |  DSI   |  NCI   | TIGER  |   DR   |    URI     |
> | :-------: | :-------: | :----: | :----: | :----: | :----: | :--------: |
> | Recall@10 |  0.4621   | 0.5593 | 0.8437 | 0.5021 | 0.4085 | **0.8929** |
>
> **Weaknesses3:** How does URI deal with zero shot items?
>
> **Response:** We propose two feasible methods. The first method is ID-independent: for new items, we only use their textual embedding and directly input them into the trained decoder to generate their code in the index. The second method is ID-dependent: we use the textual embedding to find the most similar item to the new item and use the ID embedding of the item to initialize the ID embedding of the new item. We use 10% of the items in the dataset as cold-start items, while the other items are used for normal training. The results of Recall@10 obtained are as follows. "Random" means that we randomly assign codes to cold-start items. "No Cold-Start" means that all the items are used for normal training.
>
> |     Method     | Random | ID-independent | ID-dependent | No Cold-Start |
> | :------------: | :----: | :------------: | :----------: | :-----------: |
> |    KuaiSAR     | 0.1617 |     0.1769     |  **0.1788**  |   *0.1793*    |
> | Toys and Games | 0.0509 |     0.0523     |  **0.0562**  |   *0.0573*    |
>
> From the table, we can see that using ID-dependent methods for cold-start items yields the best results, but this methods require searching, which is time-consuming. For information retrieval tasks, using only textual features can achieve good results. However, for recommendation tasks, the information gain from IDs is relatively higher. The choice of method can be tailored to the specific scenario. This is a very meaningful topic, and we will conduct more detailed experiments and analysis in the camera-ready version.
>
> **Q2:** APE is standard practice to measure cluster quality why is it a contribution in your paper. (See ECLARE)
>
> **Response:** We are sorry for the confusion. In ECLARE, the metric most similar to APE is LMI, but APE and LMI are not the same. The calculation of LMI is related to the prediction model and measures the mutual information between the true clusters and the predicted clusters. In contrast, the calculation of APE is independent of the prediction model and is only related to the index. It measures the concentration of items related to the same query within the index.

---

> > ### Comment · Reviewer_YJMb · 2024-11-23
> > **New more clarification**
> >
> > Dear Authors, I need new more clarifications
> >
> > 1) EHI is easy to replicate and all details including hyperparameters has been provided in the paper.
> > 2) URI's performance is very close to EHI on NQ-320K which validates my concerns of similarity between the two.
> > 3) From what I understand, an intuitive explanation of URI is, it uses tree by hierarchical clustering of documents and generates path of the hierarchy to reach the lead node. During training URI dynamically assign the documents to the tree path using the intermediately trained model. I would request Authors to read EHI more carefully and answer what is the difference in working principle of URI and EHI and **not just concentrate on architecture**.
> > 4) How is APE is model independent? It needs model to first index document in the cluster and then for each query it computes how densely packed documents are, this is exactly same as LMI.

---

> > > ### Author Response · Authors · 2024-11-25
> > > **Responses to Reviewer YJMb: more clarifications**
> > >
> > > Thank you for actively participating in the discussion. We will address each of your concerns in detail.
> > >
> > > **Concern3**: What is the difference in working principle of URI and EHI?
> > >
> > > **Response:** From the working principle perspective, there are two major differences between URI and EHI.
> > >
> > > + The training methods differ. URI adopts a layer-by-layer training approach, whereas EHI trains multiple layers simultaneously. When EHI trains the path embeddings of the later layers, it uses a top-k operation on the previous layers. During the initial training phase of the index, this operation can be inaccurate, potentially causing the path embeddings of the later layers to fall into local optima. In contrast, URI's layer-by-layer training ensures that the previous layers have already converged to optimal partitions when training the later layers, thus avoiding the issue of local optima caused by inaccurate partitions in the earlier layers. The difference in training methods also leads to different index representation strategies. In EHI, tokens from different layers use different representations, whereas URI uses shared representations. This not only saves memory usage but also better leverages token consistency, as mentioned in our experiment 5.3.
> > >
> > > + The loss function design differs. EHI's loss function is based on a triplet loss, which aims to widen the gap between positive and negative samples in the index to avoid index collapse while training collaborative filtering signals. URI does not require negative sampling. It directly simulates the EM algorithm proposed in the paper, using balance loss and commitment loss to assist training and prevent index collapse. Although URI's loss function also aims to bring the query and positive samples closer in the index, it is backed by the EM algorithm, providing a stronger theoretical guarantee. To further validate the reasonableness of simulating this EM algorithm, we compared the performance of directly using the EM algorithm with that of end-to-end training.
> > >
> > >   |         | Random | EM Training | End-to-End Training(URI) |
> > >   | :-----: | :----: | :---------: | :----------------------: |
> > >   | KuaiSAR | 0.0107 |   0.1199    |          0.1793          |
> > >   | NQ320K  | 0.2511 |   0.8512    |          0.8929          |
> > >
> > >   In the table above, "random" indicates training the Decoder with a randomly initialized index, while "EM training" refers to using the greedy algorithm proposed in the paper to update the index and alternately train the Decoder model. The results in the table show that the EM algorithm can also learn a relatively good index. Although it is not as effective as end-to-end training, it provides theoretical support for the loss function we designed.
> > >
> > > **Concern1**: EHI is easy to replicate and all details including hyperparameters has been provided in the paper.
> > >
> > > **Response:** We adjusted the hyperparameter in the loss function follow the original paper and successfully trained the EHI. The Recall@10 on the NQ320K and KuaiSAR datasets is shown in the table below. From the table, it can be seen that URI performs better than EHI on both datasets. This is attributed to the differences we analyzed in the third response.
> > >
> > > | Method  | Recforest |  DSI   |  NCI   | TIGER  |   DR   |  EHI   |    URI     |
> > > | :-----: | :-------: | :----: | :----: | :----: | :----: | :----: | :--------: |
> > > | NQ320K  |  0.4621   | 0.5593 | 0.8437 | 0.5021 | 0.4085 | 0.8618 | **0.8929** |
> > > | KuaiSAR |  0.0384   | 0.0976 | 0.0568 | 0.0376 | 0.0604 | 0.1309 | **0.1793** |
> > >
> > > **Concern2**: URI's performance is very close to EHI on NQ-320K which validates my concerns of similarity between the two.
> > >
> > > **Response:** Both EHI and URI train the index and retriever in an end-to-end manner, aiming to address the inconsistency between index construction and model training in two-stage methods. From the experiment results, it is evident that both methods outperform the two-stage approaches. We will include it as a competitive baseline in the camera-ready version.
> > >
> > > **Concern4**: How is APE is model independent? It needs model to first index document in the cluster and then for each query it computes how densely packed documents are, this is exactly same as LMI.
> > >
> > > **Response:** "Model Independent" means that it is independent of the model's parameters. LMI calculates the correlation between predicted classifications and true classifications in extreme classification problems. If we consider the predicted classifications in LMI as the index's classification of items, and each query's corresponding item as a classification, then the mutual information calculated in this way is indeed very similar to APE, differing only in the normalization term. Thank you for pointing this out. We will clarify this in the paper and remove it from our contributions. This does not affect our main contribution, which is an end-to-end architecture to optimize APE for indexing.

---

> > > > ### Comment · Reviewer_YJMb · 2024-11-25
> > > >
> > > > Thank you for the clarification, I have updated my scores.

---

> > > > > ### Author Response · Authors · 2024-11-26
> > > > >
> > > > > We sincerely appreciate the time you took to review our work and your positive reception of our response. Your thoughtful reconsideration of the score is truly encouraging.

---

### Official Review · Reviewer_1tQz · 2024-11-02

**Soundness:** 2
**Presentation:** 3
**Contribution:** 2
**Rating:** 6
**Confidence:** 3

**Summary:**

The paper tackles item (document) retrieval problem, i.e., the algorithm should return the indices (and scores) of most relevant items given a query. In particular, the paper focuses on generative (decoder) based subset of algorithms. Current decoder-based methods follow a two-step procedure where first the index is learned over pre-trained features (e.g., DSI) and then the model is trained to generate in this space. The paper argues that the aforementioned procedure can be sub-optimal, and it proposes URI (a unified framework for generative retrieval) that simultaneously learns the index as well as the decoder.

The model is trained in a layer-by-layer manner (in contrast to existing ones). The first layer is trained by optimizing $\mathcal{L_{E step}} + \mathcal{L_{M step}} $ - the similarity between corresponding queries and items is optimized while avoiding index collapse. The buckets are then assigned for each item via balanced assignment. First layer tokens are then passed onto second layer which is trained via an additional memory reinforcement loss term to avoid forgetting. Finally, a ranker is trained that assigns score to query item pairs on the basis of inner product between pooled embeddings.

The results are reported on KuaiSAR, Amazon Beauty and NQ-320K datasets with focus on demonstraing the benefit of URI index as compared to the baselines (DSI, NCI, TIGER etc.,).

**Strengths:**

1. The paper is decently written and easy to follow.
2. The paper reports numbers on KuaiSAR, Amazon Beauty and NQ-320K datasets and it compares against leading methods including DSI and NCI.
3. The proposed method yields good benefits when the index of baselines is replaced with the proposed URI based index as well as in terms of overall numbers. For e.g., the performance of DSI and NCI improved by ~3% on the NQ-320K datasets. The benefits are even starker on the KuaiSAR dataset where the overall performance improved by upto 14% as compared to the baselines.
4. The paper analyses the impact of URI index as compared to other popular indices including kMeans and VQ-VAE. The APE values are reported in Table 2. The benefits of layer wise training (token consistency), Adaptive weights ($\mathcal{L}_{balance\ w}$) and hard negative mining (during ranking).
5. The paper provides hyper-parameter details in 5.4.

**Weaknesses:**

1. The results section compares against various methods including DR, DSI and NCI. It shows the impact of changing the original index with URI index alongside the overall comparison. However, the overall numbers don't seem to be for the best version of respective algorithms. It is perhaps done for consistency across algorithms? For e.g., the R@20 for NQ-320K can be in range of 0.56 - 0.89 for methods including ANCE, DSI, and NCI whereas URI is 0.057 (?). Please see https://arxiv.org/pdf/2206.02743. Please clarify if I am misinterpreting something.
2. The paper is missing the training and inference times of the proposed method. These stats will provide a fair comparison against the baselines (in addition to already included theoretical complexity).

**Questions:**

1. Please look into the points above concerning end-to-end baselines and training/inference time.
2. The final scores are computed via a re-ranker that uses the inner product (on pooled query and doc embeddings). It may be worthwhile to try out a cross-encoder based re-ranker.
3. Can you please comment on scalability of URI? How does the performance change with increase in number of docs / items?

---

> ### Author Response · Authors · 2024-11-21
> **Responses to Reviewer 1tQz**
>
> Thank you for your helpful and detailed comments. We believe that your comments will make our work more robust and convincing. The detailed responses to your comments are listed as follows:
>
> **Weaknesses1:** The results on NQ-320K are not the best version of respective algorithms.
>
> **Response:** We apologize for the confusion caused by the typo error. NQ320K was mentioned only in Table 1, but the actual dataset used was Toys and Games. The cause of this error is that we initially chose NQ320K dataset, but since both it and the KuaiSAR dataset are IR tasks, we decided to replace it with the Toys and Games dataset to observe URI's performance across more tasks. All the results related are based on the Toys and Games dataset. We apologize again for our oversight in not correcting this error in Table 1. We will correct this error in the camera-ready version. To further address your concerns, we now provide the experimental results for NQ320K. Due to resource constraints, we use the Base version from the NCI paper. From the table, we can see that URI outperforms other methods on NQ320K dataset.
>
> |  Method   | Recforest |  DSI   |  NCI   | TIGER  |   DR   |    URI     |
> | :-------: | :-------: | :----: | :----: | :----: | :----: | :--------: |
> | Recall@10 |  0.4621   | 0.5593 | 0.8437 | 0.5021 | 0.4085 | **0.8929** |
>
> **Weaknesses2:** The paper is missing the training and inference time of the proposed method.
>
> **Response:** Thank you for your suggestion. We will include the efficacy analysis in the camera-ready version. The table below presents the analysis on the KuaiSAR dataset. From the table, it can be seen that the training time for URI is relatively long due to the hierarchical training process. However, this is acceptable considering its superior performance. A higher QPS (Queries Per Second) indicates shorter inference time, and both URI and DR have high QPS because they use ranker sorting, whereas other methods require an additional final layer for beam search, which is more time-consuming. In terms of space usage, both DR and URI have higher model parameters due to the presence of an item encoder.
>
> |               | RecForest |  DSI  |  NCI  | TIGER |  DR   |  URI  |
> | :-----------: | :-------: | :---: | :---: | :---: | :---: | :---: |
> | Training Time |   2.39h   | 2.16h | 2.75h | 3.03h | 4.51h | 3.27h |
> | Inference QPS |   7144    | 8412  | 8322  | 7719  | 9123  | 9556  |
> | Memory Usage  |   112M    | 112M  | 112M  | 112M  | 154M  | 154M  |
>
> **Q2:** The final scores are computed via a re-ranker that uses the inner product. It may be worthwhile to try out a cross-encoder based re-ranker.
>
> **Response:** The design of the re-ranker is not the main contribution of this paper. We chose to add an additional ranker model because our experiments showed that this approach performed better than adding extra tokens in the final layer. However, since you mentioned using the cross-attention method, we tried this approach on the sequential recommendation task for the Toys and Games dataset, and indeed, the results are better than using the dot product.
>
> |           | Inner Product | Cross Attention |
> | :-------: | :-----------: | :-------------: |
> | Recall@10 |    0.0573     |     0.0641      |
>
> **Q3:** Can you please comment on scalability of URI? How does the performance change with increase in number of docs / items?
>
> **Response:** The training of URI relies on positive query-item pairs. The more such samples there are, the better URI can capture the collaborative filtering information in the data distribution, leading to improved performance. Therefore, as the number of items increases, the effectiveness of URI improves. This is evident from the improvement observed across the three datasets mentioned in the paper, where the kuaiSAR dataset, which has the highest number of items, shows the greatest improvement with URI.

---

> > ### Comment · Reviewer_1tQz · 2024-11-23
> >
> > Thanks for the responses. Regarding scalability: please try to include theoretical computational and memory complexity to give a better idea. I have updated the score accordingly.

---

> > > ### Author Response · Authors · 2024-11-25
> > >
> > > We sincerely appreciate your time in reviewing our work and your appreciation of our response. Your thoughtful reconsideration of the score is truly encouraging. Regarding the complexity analysis of dataset scaling, we have already included this in Section 4.5 of our paper. In the camera-ready version, we will further elaborate on this part to provide more insights into the scalability of the model.

---

### Official Review · Reviewer_qpuA · 2024-11-02

**Soundness:** 3
**Presentation:** 3
**Contribution:** 3
**Rating:** 6
**Confidence:** 3

**Summary:**

This paper proposes URI (Unified framework for Retrieval and Indexing), a novel approach that unifies index construction and retrieval model training for generative retrieval systems. The key innovation is using the same Transformer decoder both as the retriever and the indexer, constructing the index simultaneously with model training rather than as a separate pre-processing step. The paper introduces Average Partition Entropy (APE) as a new metric for evaluating generative indices and provides theoretical analysis using EM algorithm to justify their approach. The authors demonstrate URI's effectiveness through experiments on three real-world datasets.

**Strengths:**

- The paper identifies clear limitations in existing two-stage approaches where index construction is separated from retrieval model training.
- The approach of using the decoder itself as an indexer is creative and original and introduction of APE as an evaluation metric is well-justified and useful
- Addresses fundamental limitations in generative retrieval systems leads to potentially applicable across various retrieval tasks
- Could influence future work in information retrieval and recommender systems

**Weaknesses:**

Limited Empirical Analysis
- While experiments are conducted on three datasets, more details about these datasets would be helpful
- Ablation studies could better isolate the impact of different components
- Comparison with more baseline methods would strengthen the evaluation

Scalability Concerns
- The paper doesn't thoroughly discuss computational complexity
- Practical considerations for large-scale deployment are not addressed
- Memory requirements for the unified approach could be significant

**Questions:**

- How does the computational complexity of URI compare to traditional two-stage approaches? Is there a significant overhead in training time or memory usage?
- How sensitive is the method to the choice of initial conditions? Does the unified training require special initialization strategies?
- Could the authors provide more details about how URI handles cold-start scenarios where new items or queries are added to the system?
- How does URI perform in scenarios with highly imbalanced data distributions? Are there any special considerations or modifications needed?
- The theoretical analysis assumes certain statistical conditions - how often are these conditions met in practice? What happens when they are not met?
- Could the authors elaborate on potential extensions of URI to handle dynamic indices that need to be updated over time?

---

> ### Author Response · Authors · 2024-11-21
> **Responses to Reviewer qpuA's Weakness & Q1**
>
> Many thanks for your helpful and detailed comments. The detailed responses to your questions are listed as follows:
>
> **Weaknesses1:** More details about these datasets would be helpful.
>
> **Response:** We have provided detailed information about the dataset in Appendix B.
>
> **Weaknesses2:** Ablation studies could better isolate the impact of different components.
>
> **Response:** We will include more ablation studies. In the loss function of URI, each term is crucial, and directly ablating any term would cause the model to fail to converge, making it hard to evaluate. Here, we will first supplement ablation study on the importance of ID features and text features. We evaluated the performance of URI using only one type of feature on the KuaiSAR and Toys datasets. The table below shows the results for Recall@10.
>
> |                | URI w/ only text feature | URI w/ only ID feature |  URI   |
> | :------------: | :----------------------: | :--------------------: | :----: |
> |    KuaiSAR     |          0.1679          |         0.1012         | 0.1793 |
> | Toys and Games |          0.0329          |         0.0491         | 0.0573 |
>
> From the table, it is evident that both ID features and text features play a crucial role in the training of URI. The importance of these features varies across different tasks. For IR tasks, text features are more important, whereas for recommendation tasks, ID features are more significant. This is a very meaningful topic, and we will conduct more detailed experiments and analysis in the camera-ready version.
>
> **Weaknesses3:** Comparison with more baseline methods would strengthen the evaluation.
>
> **Response:** We will add ASI and GenRet as two additional baselines. Both ASI[1] and GenRet[2] are designed for the Information Retrieval task. We have added a comparative experiment between these two methods and URI on the KuaiSAR dataset. The results are as follows:
>
> |           | RecForest |  DSI   |  NCI   | TIGER  |   DR   | ASI    | GenRet | URI    |
> | --------- | :-------: | :----: | :----: | :----: | :----: | ------ | ------ | ------ |
> | Recall@10 |  0.0384   | 0.0976 | 0.0568 | 0.0376 | 0.0604 | 0.0833 | 0.1021 | 0.1793 |
>
> ASI enhances the accuracy of the codebook through reconstruction, while GenRet designs a loss function to align the codes of the Query and Document, which makes them better than other baselines. However, both methods still rely on constructing indices in Euclidean space. The indexer(codebook) and the final retrieval model (Decoder) still exhibit some incompatibility, which limits their performance.
>
> [1] Yang T, Song M, Zhang Z, et al. Auto Search Indexer for End-to-End Document Retrieval
>
> [2] Sun W, Yan L, Chen Z, et al. Learning to tokenize for generative retrieval
>
> **Weaknesses4-6:** Please see the responses to Q1.
>
> **Q1:** How does the computational complexity of URI compare to two-stage approaches? Is there a significant overhead in training time or memory usage?
>
> **Response:** In Section 4.5 of the paper, we provided a complexity analysis of URI and the two-stage model. From a complexity perspective, the training time complexity of URI is 2$L$ times that of the two-stage model. Since $L$ is usually small ($L=2$ in our paper), the additional time overhead is acceptable. Moreover, URI does not require pre-training, which saves that portion of the time overhead. As for memory usage, URI adds an encoder for the item part compared to the two-stage model, which is a completely acceptable overhead.
>
> The table below presents the analysis on the KuaiSAR dataset. From the table, it can be seen that the training time for URI is relatively long due to the hierarchical training process. However, this is acceptable considering its superior performance. A higher QPS (Queries Per Second) indicates shorter inference time, and both URI and DR have high QPS because they use ranker sorting, whereas other methods require an additional final layer for beam search, which is more time-consuming. In terms of memory usage, both DR and URI have more model parameters due to the presence of an item encoder.
>
> |               | RecForest |  DSI  |  NCI  | TIGER |  DR   |  URI  |
> | :-----------: | :-------: | :---: | :---: | :---: | :---: | :---: |
> | Training Time |   2.39h   | 2.16h | 2.75h | 3.03h | 4.51h | 3.27h |
> | Inference QPS |   7144    | 8412  | 8322  | 7719  | 9123  | 9556  |
> | Memory Usage  |   112M    | 112M  | 112M  | 112M  | 154M  | 154M  |

---

> ### Author Response · Authors · 2024-11-21
> **Responses to Reviewer qpuA's Q2~Q6**
>
> **Q2:** How sensitive is the method to the choice of initial conditions? Does the unified training require special initialization strategies?
>
> **Response:** We have attempted to warm-start the model parameters. Specifically, we trained a two-stage model and then used the parameters of this model to warm-start URI. We found that this approach can accelerate the convergence of the model, but the final performance after convergence is not as good as training from scratch. The table below shows a comparison of the results on KuaiSAR between cold-start and warm-start.
>
> |               | Warm-start URI | Cold-start URI |
> | :-----------: | :------------: | :------------: |
> | Training Time |     1.32h      |     3.27h      |
> |   Recall@10   |     0.1511     |     0.1793     |
>
> **Q3:** Could the authors provide more details about how URI handles cold-start scenarios where new items or queries are added to the system?
>
> **Response:** We propose two feasible methods for cold start items. The first method is ID-independent: for new items, we only use their textual embedding and directly input them into the trained decoder to generate their code in the index. The second method is ID-dependent: we use the textual embedding to find the most similar item to the new item and use the ID embedding of the item to initialize the ID embedding of the new item. We use 10% of the items in the dataset as cold-start items, while the other items are used for normal training. The results of Recall@10 obtained are as follows. "Random" means that we randomly assign codes to cold-start items. "No Cold-Start" means that all the items are used for normal training.
>
> |     Method     | Random | ID-independent | ID-dependent | No Cold-Start Item |
> | :------------: | :----: | :------------: | :----------: | :----------------: |
> |    KuaiSAR     | 0.1617 |     0.1769     |  **0.1788**  |      *0.1793*      |
> | Toys and Games | 0.0509 |     0.0523     |  **0.0562**  |      *0.0573*      |
>
> From the table, we can see that using ID-dependent methods for cold-start items yields the best results, but this method require searching, which is time-consuming. For information retrieval tasks, using only textual features can achieve good results. However, for recommendation tasks, the information gain from IDs is relatively higher. We obtain conclusions similar to those in our response to Weakness2. The choice of method can be tailored to the specific scenario.
>
> **Q4:** How does URI perform in scenarios with highly imbalanced data distributions? Are there any special considerations or modifications needed?
>
> **Response:** The biggest impact of imbalanced data distribution is that it can lead to an imbalanced index, which ultimately affects performance. However, during training, we applied both soft and hard balancing constraints to the index. The soft balancing constraint refers to a balance loss term included in our loss function, while the hard balancing constraint refers to limiting the load of each token node during item assignment. Both of these measures effectively prevent the issue of index imbalance.
>
> **Q5:** How often are the conditions in Theorem met in practice? What happens when they are not met?
>
> **Response:** In fact, the conditions of Theorem 1 can be easily satisfied, as we have demonstrated in Appendix B. For a tolerance level of 0.95, all three datasets used in the paper meet the conditions of Theorem 1. This indicates that the greedy algorithm described in Theorem 1 can correctly allocate at least 95% of the items for these datasets. When the conditions are not met, the greedy algorithm described in Theorem 1 may lead to inaccurate item allocation. However, this is not the final method in our paper. We use an end-to-end training approach. Although it simulates the greedy algorithm, fundamentally it captures the collaborative filtering signals in the dataset, which allows us to avoid bad allocation.
>
> **Q6:** Could the authors elaborate on potential extensions of URI to handle dynamic indices that need to be updated over time?
>
> **Response:** In our response to Q3, we provided methods for handling cold-start items, which addresses the issue of dynamic index insertion. For the deletion of outdated items, entries for those items can be directly removed from their respective buckets. If there is a large influx of new data, such as user-item interaction data, the index needs to be retrained. Warm-starting the training with existing parameters can lead to rapid convergence of the index, as it essentially only involves adjusting the allocation of a part of items.

---

### Official Review · Reviewer_esfx · 2024-11-04

**Soundness:** 2
**Presentation:** 2
**Contribution:** 2
**Rating:** 6
**Confidence:** 3

**Summary:**

This paper introduces URI (Unified Retrieval and Indexing), a framework for generative retrieval that integrates index construction with Transformer Decoder training. URI enhances consistency between the index and retrieval model. The authors also introduce Average Partition Entropy (APE), a model-independent metric for evaluating generative indices after index construction. Targeting this metric, they also propose an optimization algorithm with a theoretical explanation. Experiments confirm the effectiveness of the proposed method.

**Strengths:**

- The motivation is clear. The authors argue that recent methods separate index construction from retrieval  in traditional and generative models, which limits the overall performance. Thus, this paper attempts to unify index construction and retrieval.
- This paper provides a theoretical guarantee of the proposed greedy algorithm.
- The experiments on multiple benchmark datasets validate the effectiveness of the proposed method compared with the baseline models.

**Weaknesses:**

- The authors overlook recent retrieval work such as [1,2]. The authors are encouraged to discuss them in the related work.
- Since the final goal of the proposed method has many hyper-parameters in (11), more sensitivity analysis also is crucial.
- Theorem 1 lacks the corresponding empirical results, so more evidence is needed.
- Including both index construction and retrieval in the unified framework may increase computation and storage overhead, affecting efficiency. Therefore, an efficacy analysis (e.g., memory usage, training/inference time, and FLOPs) is necessary.

[1] Improved Diversity-Promoting Collaborative Metric Learning for Recommendation. IEEE TPAMI, 2024.

[2]Hierarchical latent relation modeling for collaborative metric learning,”inRecSys,2021

**Questions:**

Please see the above weakness.

---

> ### Author Response · Authors · 2024-11-21
> **Responses to Reviewer esfx**
>
> Thank you for your insightful suggestions! We appreciate your encouragement and provide detailed responses to your comments below:
>
> **Weaknesses1:** The authors overlook recent retrieval work such as HLR and DPCML. The authors are encouraged to discuss them in the related work.
>
> **Response:** Thank you for your suggestion! Both of these works are based on Collaborative Metric Learning. HLR proposes a hierarchical CML model that jointly captures latent user-item and item-item relationships from implicit data. DPCML proposes a method called Diversity-Promoting Collaborative Metric Learning, which considers the commonly ignored minority interests of users. We will include a discussion of these two works in the related work section of the camera-ready version.
>
> **Weaknesses2:** Since the final goal of the proposed method has many hyper-parameters in (11), more sensitivity analysis of hyper-parameters is crucial.
>
> **Response:** In Loss Function 11, there are only two parameters, $k$ and $l$. Here, $k$ represents the width of the index, and $l$ represents the current layer being trained, with $l$ having a maximum value of $L$, which is the depth of the index. We have already provided a sensitivity analysis of these two parameters in Section 5.4 of the paper. The other symbols are not hyperparameters, and their meanings are explained in the sections where they are introduced. Additionally, we have further summarized the meaning of each symbol in the Appendix.
>
> **Weaknesses3:** Theorem 1 lacks the corresponding empirical results, so more evidence is needed.
>
> **Response:** The conditions of Theorem 1 can be easily satisfied, as we have demonstrated in Appendix B. For a tolerance level of 0.95, all three datasets used in the paper meet the conditions of Theorem 1. This indicates that the greedy algorithm described in Theorem 1 can correctly allocate at least 95% of the items in these datasets.
>
> **Weaknesses4:** An efficacy analysis (e.g., memory usage, training/inference time, and FLOPs) is necessary.
>
> **Response:** Thank you for your suggestion! We will include the efficacy analysis in the camera-ready version. The table below presents the analysis on the KuaiSAR dataset. From the table, it can be seen that the training time for URI is relatively long due to the hierarchical training process. However, this is acceptable considering its superior performance. A higher QPS (Queries Per Second) indicates shorter inference time, and both URI and DR have high QPS because they use ranker sorting, whereas other methods require an additional final layer for beam search, which is more time-consuming. In terms of memory usage, both DR and URI have more model parameters due to the presence of an item encoder.
>
> |               | RecForest |  DSI  |  NCI  | TIGER |  DR   |  URI  |
> | :-----------: | :-------: | :---: | :---: | :---: | :---: | :---: |
> | Training Time |   2.39h   | 2.16h | 2.75h | 3.03h | 4.51h | 3.27h |
> | Inference QPS |   7144    | 8412  | 8322  | 7719  | 9123  | 9556  |
> | Memory Usage  |   112M    | 112M  | 112M  | 112M  | 154M  | 154M  |

---

### Official Review · Reviewer_Qbkc · 2024-11-04

**Soundness:** 2
**Presentation:** 1
**Contribution:** 2
**Rating:** 6
**Confidence:** 4

**Summary:**

This paper proposes URI, a variant of generative retrieval methods which learns the retrieval model and index construction jointly. Besides, this paper also proposes averaged partition entropy (APE) as a measure for index construction quality. In URI, the same decoder is used to generate tokens for both query and items and is trained in a using an EM algorithm. In E-step, the model is optimized through minimizing the cross entropy loss.  In M-step, To avoid model collapse while minimizing APE, two loss functions apart from the cross entropy loss are introduced to encourage that the token distribution w.r.t. each item follows a peak distribution while the expected token distribution w.r.t. the whole item space follows a uniform distribution. Finally, an additional dual encoder model is trained to identify the topk items as the final retrieved result. Experiments on KuaiSAR and Amazon Beauty/Toys/Games demonstrates the effectiveness of URI compared to other generative retrieval methods.

**Strengths:**

1. Learning to construct index jointly with retrieval models is an important problem for generative retrieval. The proposed method is well motivated.
2. Performance is impressive, especially on the KuaiSAR dataset.

**Weaknesses:**

1. Writing can be further improved, especially the experiment section. The difference between URI and GR methods w/ URI index is unclear in Table 1. The analysis of token consistency is hard to understand and it will be better to demonstrate their differences to original URI through notations.
2. Insufficient analysis and comparison on other generative methods (e.g., ASI [1] and GenRet [2] ) that learns both retrieval models and indexing jointly.
3. The assumption of Theorem 1 seems very strong and there lacks empirical analysis on real-world dataset to verify the rationality.
4. The comparison in experiments is not unfair. The candidate size of URI is larger than other GR methods which requires the mapping between the token representation and the item is bijective while URI allows each leaf node contains more than 1 items. Besides, URI is equipped with an additional ranker.

[1] Yang T, Song M, Zhang Z, et al. Auto Search Indexer for End-to-End Document Retrieval[C]//Findings of the Association for Computational Linguistics: EMNLP 2023. 2023: 6955-6970.
[2] Sun W, Yan L, Chen Z, et al. Learning to tokenize for generative retrieval[J]. Advances in Neural Information Processing Systems, 2024, 36.

**Questions:**

My major concerns are listed in the weakness section.

---

> ### Author Response · Authors · 2024-11-21
> **Responses to Reviewer Qbkc**
>
> Many thanks for your helpful and detailed comments. We believe that your comments will make our work more robust and convincing. The detailed responses to your comments are listed as follows:
>
> **Weaknesses1:** Writing can be further improved: 1.difference between URI and GR methods w/ URI index; 2. The analysis of token consistency
>
> **Response:** For the two points you mentioned, we will add the following explanations to the paper:
>
> 1. “GR methods w/ URI index” indicates that we retain the model structure and training methods of the GR methods, but only change the index. For example, for NCI, we use the PAWA network as described in the original paper. However, the index is no longer obtained through hierarchical clustering; instead, it is replaced with the URI-trained index.
> 2. “Token consistency” refers to the consistency of the same code across different levels. We will add symbolic representations to clarify this. For instance, in an index with 2 layers, the codes for items might be $[3^1, 2^2]$, $[4^1, 2^2]$, and $[2^1, 3^2]$, where the superscripts indicate the layer number. “Token consistency” has two meanings: The first meaning is that the same code across different levels has similarity, such as $2^2$ and $2^1$ in $[3^1, 2^2]$ and $[2^1, 3^2]$. The second meaning is that the same code under different parent nodes also has similarity, such as $2^2$ in $[3^1, 2^2]$ and $[4^1, 2^2]$ .
>
> For other parts of the experiment, we will also use symbols or diagrams to make the presentation clearer. If you have any specific suggestions for modifications, please let us know.
>
> **Weaknesses2:** Insufficient analysis and comparison on other generative methods (e.g., ASI and GenRet)
>
> **Response:** Both ASI and GenRet are designed for the Information Retrieval task. We have conducted a comparative experiment between these two methods and URI on the KuaiSAR and NQ320K dataset. The results of Recall@10 are as follows:
>
> |         | RecForest |  DSI   |  NCI   | TIGER  |   DR   | ASI    | GenRet | URI        |
> | ------- | :-------: | :----: | :----: | :----: | :----: | ------ | ------ | ---------- |
> | KuaiSAR |  0.0384   | 0.0976 | 0.0568 | 0.0376 | 0.0604 | 0.0833 | 0.1021 | **0.1793** |
> | NQ320K  |  0.4621   | 0.5593 | 0.8437 | 0.5021 | 0.4085 | 0.8788 | 0.8667 | **0.8929** |
>
> ASI enhances the accuracy of the codebook through reconstruction, while GenRet designs a loss function to align the codes of the Query and Document, making them better than other baselines. However, both methods still rely on constructing indices in Euclidean space. The indexer (codebook) and the final retrieval model (Decoder) still exhibit some incompatibility, limiting their performance. We will include these two baselines in the camera-ready version and provide experiments on additional datasets.
>
> **Weaknesses3:** The assumption of Theorem 1 seems very strong and there lacks empirical analysis on real-world dataset.
>
> **Response:** The conditions of Theorem 1 can be easily satisfied, as demonstrated in Appendix B. For a tolerance level of 0.95, all three datasets used in the paper meet the conditions of Theorem 1. This indicates that the greedy algorithm described in Theorem 1 can correctly allocate at least 95% of the items in these datasets.
>
> **Weaknesses4:** The comparison in experiments is not unfair. The candidate size of URI is large and URI is equipped with an additional ranker.
>
> **Response:** We apologize for the misunderstanding. In fact, the candidate size is the same for all methods, as the beam size is uniformly set to 20. For URI and DR, this means selecting the top 20 buckets, while for other methods, it means selecting the top 20 parent nodes. Regardless of whether they are buckets or parent nodes, the candidate items attached are the same. The difference lies in the ranking process: URI and DR use an additional ranker, while other models still use the probabilities output by the Decoder for sorting. We use an additional ranker because it yields better results. Additionally, in Table 1, we compared the performance of the model structures of all methods on both the original index and the URI index. The results show a significant improvement in performance on the URI index. This phenomenon is unrelated to the ranker and demonstrates the inherent superiority of the URI index.

---

> ### Author Response · Authors · 2024-11-25
> **Seeking Reviewer Feedback: Kind Reminder for the Final Stage of Author-Reviewer Discussion**
>
> Dear Reviewer,
>
> We would like to express our sincere gratitude for all the help you have provided in reviewing our work. Your feedback has been invaluable in helping us improve our submission. Since the deadline for the author-reviewer discussion is approaching and we have yet to receive your further feedback, we hope to kindly request your help once again. We would greatly appreciate it if we could have further discussions with you to ensure that our submission meets the high standards of ICLR.
>
> Best regards,
>
> Authors

---

> > ### Comment · Reviewer_Qbkc · 2024-11-26
> >
> > Thanks for your response. I have raised my score

---

> > > ### Author Response · Authors · 2024-11-30
> > >
> > > We are truly grateful for the time you dedicated to reviewing our work and for your positive reception of our response. Your thoughtful reconsideration of the score is highly encouraging.

---

### Author Response · Authors · 2024-12-03

We sincerely appreciate all the reviewers for their efforts during the review process and for providing valuable suggestions. Their active participation in discussions helped us address the issues in our paper. We are very pleased and grateful that all the reviewers ultimately recommended acceptance. They recognized that our work is well-motivated (Reviewer Qbkc, esfx, qpuA), the proposed methods and theoretical guarantees are effective (Reviewer esfx, qpuA, 1tQz), and the experimental results are impressive (Reviewer Qbkc, esfx, 1tQz, YJMb). We have made every effort to address each reviewer's concerns and have conducted the requested experiments. Once again, we thank all the reviewers for their assistance in improving our work.

---

### Meta-Review · Area_Chair_JRck · 2024-12-19

**Metareview:**

This paper introduces a framework for generative retrieval that combines index construction with Transformer Decoder training. Initially, the paper received mixed reviews; however, after extensive discussions between authors and reviewers, all reviewers have now expressed a positive stance. Therefore, I am pleased to recommend its acceptance.  Nonetheless, the authors must include all the discussions in the final version of their paper.

**Additional Comments On Reviewer Discussion:**

- Reviewer `Qbkc`: raised the score to 6 after seeing the additional experimental analysis the authors provided in the rebuttal phase

- Reviewer `esfx`: weakly positive (6). The authors have addressed the major issues, especially the validation of the theorem and the efficiency analysis.

-  Reviewer `qpuA`: weakly positive (6).  The authors have addressed the major issues, especially the ablation studies and baselines.

- Reviewer `1tQz` raised the score to 6 and asked the authors to include theoretical computational and memory complexity. This is not a big issue since the paper has partially included some details.

- Reviewer `YJMb`:  raised the score to 6 and appreciated clarifying the author's response, especially the relation between URI and EHI.

All reviewers are now positive toward this paper, and I therefore recommend accepting this paper.

---

### Decision · Program_Chairs · 2025-01-22

Accept (Poster)